# DUAL ALGORITHMIC REASONING

**Danilo Numeroso**
University of Pisa
danilo.numeroso@phd.unipi.it

**Davide Bacciu**
University of Pisa
davide.bacciu@unipi.it

**Petar Veličković**
DeepMind
petarv@deepmind.com

## ABSTRACT

Neural Algorithmic Reasoning is an emerging area of machine learning which seeks to infuse algorithmic computation in neural networks, typically by training neural models to approximate steps of classical algorithms. In this context, much of the current work has focused on learning reachability and shortest path graph algorithms, showing that joint learning on similar algorithms is beneficial for generalisation. However, when targeting more complex problems, such "similar" algorithms become more difficult to find. Here, we propose to learn algorithms by exploiting *duality* of the underlying algorithmic problem. Many algorithms solve optimisation problems. We demonstrate that simultaneously learning the dual definition of these optimisation problems in algorithmic learning allows for better learning and qualitatively better solutions. Specifically, we exploit the max-flow min-cut theorem to simultaneously learn these two algorithms over synthetically generated graphs, demonstrating the effectiveness of the proposed approach. We then validate the real-world utility of our dual algorithmic reasoner by deploying it on a challenging brain vessel classification task, which likely depends on the vessels' flow properties. We demonstrate a clear performance gain when using our model within such a context, and empirically show that learning the max-flow and min-cut algorithms together is critical for achieving such a result.

## 1 INTRODUCTION

Learning to perform algorithmic-like computation is a core problem in machine learning that has been widely studied from different perspectives, such as learning to reason (Khardon & Roth, 1997), program interpreters (Reed & De Freitas, 2015) and automated theorem proving (Rocktäschel & Riedel, 2017). As a matter of fact, enabling reasoning capabilities of neural networks might drastically increase *generalisation*, i.e. the ability of neural networks to generalise beyond the support of the training data, which is usually a difficult challenge with current neural models (Neyshabur et al., 2017). Neural Algorithmic Reasoning (Velickovic & Blundell, 2021) is a recent response to this long-standing question, attempting to train neural networks to exhibit some degrees of *algorithmic reasoning* by learning to execute classical algorithms. Arguably, algorithms are designed to be general, being able to be executed and return "optimal" answers for any inputs that meet a set of strict pre-conditions. On the other hand, neural networks are more flexible, i.e. can adapt to virtually any input. Hence, the fundamental question is whether neural models may inherit some of the positive algorithmic properties and use them to solve potentially challenging real-world problems.

Historically, learning algorithms has been tackled as a simple supervised learning problem (Graves et al., 2014; Vinyals et al., 2015), i.e. by learning an input-output mapping, or through the lens of reinforcement learning (Kool et al., 2019). However, more recent works build upon the notion of *algorithmic alignment* (Xu et al., 2020) stating that there must be an "alignment" between the learning model structure and the target algorithm in order to ease optimisation. Much focus has been placed on Graph Neural Networks (GNNs) (Bacciu et al., 2020) learning graph algorithms, i.e Bellman-Ford (Bellman, 1958). Velickovic et al. (2020b) show that it is indeed possible to train GNNs to execute classical graph algorithms. Furthermore, they show that optimisation must occur

on all the intermediate steps of a graph algorithm, letting the network learn to replicate step-wise transformations of the input rather than learning a map from graphs to desired outputs. Since then, algorithmic reasoning has been applied with success in reinforcement learning (Deac et al., 2021), physics simulation (Velickovic et al., 2021) and bipartite matching (Georgiev & Lió, 2020).

Moreover, Xhonneux et al. (2021) verify the importance of training on multiple "similar" algorithms at once (*multi-task learning*). The rationale is that many classical algorithms share sub-routines, i.e. Bellman-Ford and Breadth-First Search (BFS), which help the network learn more effectively and be able to transfer knowledge among the target algorithms. Ibarz et al. (2022) expand on this concept by building a generalist neural algorithmic learner that can effectively learn to execute even a set of unrelated algorithms. However, learning some specific algorithms might require learning of very specific properties of the input data, for which multi-task learning may not help. For instance, learning the *Ford-Fulkerson algorithm* (Ford & Fulkerson, 1956) for *maximum flow* entails learning to identify the set of *critical* (bottleneck) edges of the flow network, i.e. edges for which a decrease in the *edge capacity* would decrease the maximum flow. Furthermore, in the single-task regime, i.e. when we are interested in learning only one single algorithm, relying on multi-task learning can unnecessarily increase the computational burden on the training phase.

Motivated by these requirements, we seek alternative learning setups to alleviate the need for training on multiple algorithms and enable better reasoning abilities of our algorithmic reasoners. We find a potentially good candidate in the *duality* information of the target algorithmic problem. The concept of duality fundamentally enables an algorithmic problem, e.g. linear program, to be viewed from two perspectives, that of a *primal* and a *dual* problem. These two problems are usually complementary, i.e. the solution of one might lead to the solution of the other. Hence, we propose to incorporate duality information directly in the learning model both as an additional supervision signal and input feature (by letting the network reuse its dual prediction in subsequent steps of the algorithm), an approach we refer to as Dual Algorithmic Reasoning (DAR). To the best of our knowledge, there exists no prior work targeting the usage of duality in algorithmic reasoning. We show that by training an *algorithmic reasoner* on both learning of an algorithm and optimisation of the dual problem we can relax the assumption of having multiple algorithms to train on while retaining all the benefits of multi-task learning. We demonstrate clear performance gain in both synthetically generated algorithmic tasks and real-world predictive graph learning problems.

## 2    PROBLEM STATEMENT

We study the problem of neural algorithmic reasoning on graphs. Specifically, we target learning of graph algorithms $A : \mathbb{G} \to \mathbb{Y}$ that take in graph-structured inputs $G = (V, E, \boldsymbol{x}_i, \boldsymbol{e}_{ij})$, with $V$ being the set of nodes and $E$ the set of edges with node features $\boldsymbol{x}_i$ and edge features $\boldsymbol{e}_{ij}$, and compute a desired output $\boldsymbol{y} \in \mathbb{Y}$. Usually, the output space of an algorithm $A$ depends on its scope. In the most general cases, it can either be $\mathbb{R}^{|V|}$ (node-level output), $\mathbb{R}^{|V| \times |V|}$ (edge-level output) or $\mathbb{R}$ (graph-level output). We mainly consider the class of algorithms outputting node-level and edge-level outputs, which includes many of the most well-known graph problems, e.g. reachability, shortest path and maximum flow. From a neural algorithmic reasoning perspective, we are particularly interested in learning a sequence of transformations (*steps of the algorithm*). Hence, we consider a sequence of graphs $\{G^{(0)}, \dots, G^{(T-1)}\}$ where each element represents the intermediate state of the target algorithm we aim to learn. At each step $t$ we have access to intermediate node and edge features, i.e. $\boldsymbol{x}_i^{(t)}, \boldsymbol{e}_{ij}^{(t)}$, called *hints* as well as intermediate targets $\boldsymbol{y}^{(t)}$. As it is common in classical algorithms, some of the intermediate targets may be used as node/edge features in the subsequent step of the algorithm. Such hints are thus incorporated in training as additional features/learning targets, effectively learning the whole sequence of steps (*algorithm trajectory*).

In particular, we focus on learning *maximum flow* via the neural execution of the *Ford-Fulkerson algorithm*. Differently from Georgiev & Lió (2020), who learn Ford-Fulkerson to find the independent set of edges in bipartite graphs, we aim to learn Ford-Fulkerson for general graphs. We report the pseudo-code of Ford-Fulkerson in the appendix. Ford-Fulkerson poses two key challenges: (i) it comprises two sub-routines, i.e. finding augmenting paths from $s$ to $t$, and updating the flow assignment $\boldsymbol{F}^{(t)} \in \mathbb{R}^{|V| \times |V|}$ at each step $t$; (ii) $\boldsymbol{F}$ must obey a set of strict constraints, namely the *edge-capacity constraint* and *conservation of flows*. The former states that a scalar value $c_{ij}$

(capacity) is assigned to every $(i, j) \in E$ and $\boldsymbol{F}$ must satisfy:

$$\forall (i, j) \in E \; . \; \boldsymbol{F}_{ij} \leq c_{ij}, \tag{1}$$

i.e. flow assignment to an edge must not exceed its capacity. The latter states that the assignment needs to satisfy:

$$\forall i \in V \setminus \{s, t\} : \sum_{(i,j) \in E} \boldsymbol{F}_{ij} + \sum_{(j,i) \in E} \boldsymbol{F}_{ji} = 0 \quad \wedge \sum_{(s,j) \in E} \boldsymbol{F}_{sj} = - \sum_{(j,t) \in E} \boldsymbol{F}_{jt} \tag{2}$$

i.e. the flow sent out from the source is not lost nor created by intermediate nodes. This also leads to $\boldsymbol{F} = -\boldsymbol{F}^T$, i.e. antisymmetry. An optimal solution $\boldsymbol{F}^*$ to the max flow problem is the one maximising the total flow in the network, i.e. $\sum_{(s,j) \in E} \boldsymbol{F}_{sj}$. We show how we address both challenge (i) and (ii) directly in the model architecture, through the concept of *algorithmic alignment* and by carefully adjusting and rescaling the model predictions.

## 3 METHODOLOGY

### 3.1 LEVERAGING DUALITY IN ALGORITHMIC REASONING

We leverage the concept of *duality* when learning to neurally execute classical graph algorithms. In particular, most of the problems solved by classical algorithms (including maximum flow) can be expressed in the form of constrained optimisation problems such as *linear programming* or *integer linear programming*. In mathematical optimisation, the duality principle ensures that any optimisation problem may be viewed from two perspectives: the "direct" interpretation (called the *primal* problem) and the *dual* problem, which is usually derived from the *Lagrangian* of the primal problem. The duality principle ensures that the solutions of the two problems are either linked by an upper-bound/lower-bound relation (*weak duality*) or equal (*strong duality*) (Boyd et al., 2004). Hence, the two problems are interconnected.

In the context of neural algorithmic reasoning, we identify several reasons why primal-dual information might be useful to consider. First, by incorporating *primal-dual* objectives, we let the network reason on the task from two different and complementary perspectives. This can substantially simplify learning of algorithms which require identifying and reasoning on properties which are not explicitly encoded in input data. For instance, to effectively solve max-flow problems, the network needs the ability to identify and reason on *critical edges*. By the *max-flow min-cut theorem* (Ford Jr & Fulkerson, 2015), this set of edges corresponds to the *minimum cut*, i.e. dual problem, that separates the source node $s$ from the sink $t$. Hence, correctly identifying the minimum cut is highly relevant for producing a relevant max-flow solution.

Second, being able to output a better step-wise solution means that there is less chance for error propagation throughout the trajectory of the neurally executed algorithm. This is especially true for more complex algorithms, such as Ford-Fulkerson, consisting of multiple interlocked sub-routines. There, an imprecise approximation of one sub-routine can negatively cascade on the results of the following ones. Finally, learning jointly on the primal-dual can be seen as an instance of *multi-task learning*, but relaxing the assumption of having multiple algorithms to train on.

In the following, we study dual algorithmic reasoning on the max-flow primal complemented with min-cut dual information. Note that graph neural networks have been formally proven to be able to learn minimum cut, even under uninformative input features (Fereydounian et al., 2022). This also implies that solving min-cut can be a useful "milestone" for a network learning to solve max-flow.

### 3.2 ARCHITECTURE

We rely on the neural algorithmic reasoning blueprint (Velickovic & Blundell, 2021), building on the encode-process-decode framework (Hamrick et al., 2018). The abstract architecture of the Dual Algorithmic Reasoner (DAR) is depicted in Figure 1 for the Ford-Fulkerson algorithm. Since the latter is composed of two sub-routines, we introduce two processors to align neural execution with the dynamics of the algorithm. The first processor $P_{BF}$ learns to retrieve augmenting paths, while $P_F$ learns to perform flow-update operations $\boldsymbol{F}^{(t)}$. Both $P_{BF}$ and $P_F$ are implemented as graph

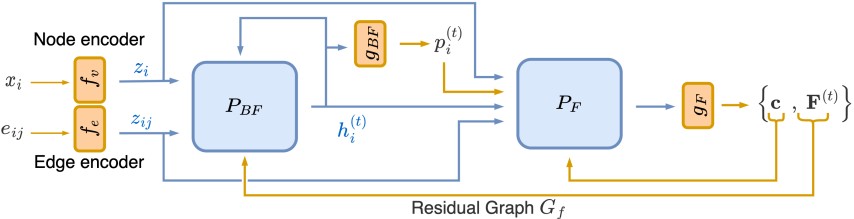

Figure 1: High-level architecture of the Dual Algorithmic Reasoner (DAR) for the Ford-Fulkerson algorithm. Refer to the text for a comprehensive explanation.

networks with Message-Passing Neural Network (MPNN) convolution (Gilmer et al., 2017):

$$\boldsymbol{h}_i^{(t+1)} = \psi_\theta \left( \boldsymbol{h}_i^{(t)}, \bigoplus_{(j,i)\in E} \phi_\theta \left( \boldsymbol{h}_i^{(t)}, \boldsymbol{h}_j^{(t)}, \boldsymbol{e}_{ij}^{(t)} \right) \right), \tag{3}$$

where $\psi_\theta$ and $\phi_\theta$ are neural networks with ReLU activations and $\bigoplus$ is a permutation-invariant function, i.e. summation, mean or max.

Intuitively, the encode-process-decode architecture allows *decoupling* learning of the algorithmic steps from the use of specific input features. Through the learned processor, the algorithm can be neurally executed on a latent-space which is a learnt representation of the input features required by the original algorithm. We will show how we can exploit this property to perform steps of the Ford-Fulkerson algorithm even with missing input features.

More in detail, the DAR computational flow comprises two *linear* encoders, $f_v$ and $f_e$, which are applied respectively to node features $\boldsymbol{x}_i^{(t)}$ and edge features $\boldsymbol{e}_{ij}^{(t)}$ to produce encoded node-level and edge-level features:

$$\boldsymbol{Z}_V^{(t)} = \{\boldsymbol{z}_i^{(t)} = f_v(\boldsymbol{x}_i^{(t-1)}) \mid \forall i \in V\} \quad , \quad \boldsymbol{Z}_E^{(t)} = \{\boldsymbol{z}_{ij}^{(t)} = f_e(\boldsymbol{e}_{ij}^{(t-1)}) \mid \forall (i,j) \in E\}.$$

These encoded representations are used as inputs for the processor network $P_{BF}$ which computes the latent node representations $\boldsymbol{H}^{(t)}$ as:

$$\boldsymbol{H}^{(t)} = P_{BF}(\boldsymbol{Z}_V^{(t)}, \boldsymbol{Z}_E^{(t)}, \boldsymbol{H}^{(t-1)})$$

with $\boldsymbol{H}^{(0)} = \{\boldsymbol{0} \mid \forall i \in V\}$. In our DAR instance, this processor performs Bellman-Ford steps to retrieve the shortest augmenting path from $s$ to $t$, following Georgiev & Lió (2020). $\boldsymbol{H}^{(t)}$ is then passed to a decoder network $g$ producing the augmenting path $\boldsymbol{p}^{(t)}$:

$$p_i^{(t)} = g_{BF}(\boldsymbol{z}_i^{(t)}, \boldsymbol{h}_i^{(t)}). \tag{4}$$

The augmenting path is represented as a vector of predecessors for all nodes in the graph, i.e. each entry $p_i^{(t)}$ is a pointer to another node $j$ in the graph. This way, we are able to reconstruct a path from any node (included $t$) back to the source node $s$. The augmenting path $\boldsymbol{p}^{(t)}$ is then passed to $P_F$ as an input feature. The target quantities of the algorithm, i.e. flow assignment $\boldsymbol{F}$ and minimum cut $\boldsymbol{c}$, are finally predicted as:

$$\{\boldsymbol{F}^{(t)}, \boldsymbol{c}\} = g_F\big(P_F(\boldsymbol{Z}_V^{(t)} \cup \{\boldsymbol{p}^{(t)}\}, \boldsymbol{Z}_E^{(t)}, \boldsymbol{H}^{(t)})\big).$$

W.l.o.g. we choose to represent the minimum $s$-$t$ cut $\boldsymbol{c}$ as node-level features, where $c_i = 0$ indicates that $i$ is in the cluster of nodes of $s$, and $c_i = 1$ otherwise. Note that the minimum $s$-$t$ cut includes all edges $(i,j)$ for which $c_i = 0$ and $c_j = 1$. Furthermore, $\boldsymbol{F}^{(t)}$ is reused as an input feature in the next step of the algorithm ($\boldsymbol{F}^{(0)} = \boldsymbol{0}$).

We pay additional attention to the prediction of the flow assignment matrix $\boldsymbol{F}^{(t)}$, in order to be compliant with the maximum flow problem constraints described in subsection 3.1. In particular, we transform $\boldsymbol{F}$ to ensure compliance with anti-symmetry, i.e. $\boldsymbol{F}' = \boldsymbol{F} - \boldsymbol{F}^T$. To satisfy edge-capacity constraints we further rescale the matrix according to the hyperbolic tangent and the actual value of the capacity $c_{ij}$ for each $(i,j) \in E$, as such:

$$\boldsymbol{F} = \tanh(\boldsymbol{F}) \odot C, \tag{5}$$

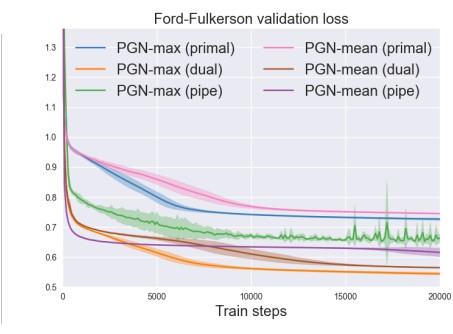
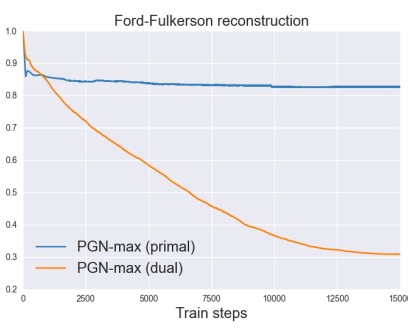

(a) Ford-Fulkerson validation loss

(b) Ford-Fulkerson reconstruction loss

Figure 2: **(a)** Ford-Fulkerson validation loss on synthetic data for PGNs. **(b)** normalised loss curve of reconstructing Ford-Fulkerson with new encoders for $l_{ij}, d_{ij}, \rho_{ij}$, with both the primal and dual PGN-max. It applies to BVG data.

where $C_{i,j} = c_{ij}$ for all edges in the graph. We note that this only satisfies the box constraint on the edge capacities, however the conservation of flows might still be violated, i.e. nodes in the path from the source to the target may either retain some amount of in-flow (sending out less than what is received) or vice versa. To address this last constraint, we simulate the entire neural algorithm until termination and apply a corrective procedure in order to correct all the flow conservation error. We report the pseudocode of this procedure in the Appendix, along with additional details and results.

## 4 EXPERIMENTS

To assess the benefits of the dual algorithmic reasoning approach, we test the learning model in two specific scenarios. First, we train and test the DAR pipeline on synthetic-generated graphs, to evaluate the benefits in the key task of algorithmic learning (section 4.1). Then, to evaluate the generality of the model we test it on a real-world graph learning task. Specifically, we compare our model with several graph learning baselines on a biologically relevant *vessel* classification task (Paetzold et al., 2021), comprehending large-scale vessel graphs (section 4.2). We stress that our neural reasoners are not further re-trained on real-world data, thus forcing the model to use the algorithmic knowledge attained on synthetic data to solve the new task.

### 4.1 SYNTHETIC GRAPHS

**Data generation**  We consider two different families of graphs: (i) *2-community* graphs, in which communities are sampled from the *Erdős–Rényi* distributions with probability 0.75 and their nodes are interconnected with probability 0.05; (ii) *bipartite* graphs. To thoroughly assess the generalisation capabilities of our algorithmic reasoners, we exclusively trained all models on small *2-community* graphs and tested on 4x larger *2-community* graphs (*out-of-distribution*) and 4x larger *bipartite* graphs (*out-of-family*). We highlight that bipartite graphs are solely used for testing purposes and no further training occurs on them. To generate train, validation and test sets we follow the standard CLRS benchmark (Veličković et al., 2022) setup. Specifically, we sample 1000 *2-community* training graphs with 16 nodes each. The validation set is used to assess in-distribution performance, thus comprising 128 *2-community* graphs with still 16 nodes. To assess out-of-distribution and out-of-family generalisation we consider respectively 128 test 2-community samples and 128 bipartite samples, both of size of 64 nodes. Furthermore, we generate data of all intermediate steps of the Ford-Fulkerson algorithm to be used as *hints* and additional training targets, in order to train the network on all intermediate data manipulations. Algorithm features are once again generated following the CLRS-30 standard and they comprise: (i) *inputs*: source node $s$, sink node $t$, edge-capacity matrix $C \in \mathbb{N}^{|V| \times |V|}$ and additional weights $W \in [0,1]^{|V| \times |V|}$ for the Bellman-Ford processor; (ii) *hints* (algorithm steps): augmenting paths $\boldsymbol{p}^{(t)}$ and intermediate flow assignments $\boldsymbol{F}^{(t)}$; (iii) *outputs* (learning targets): final flow matrix $\boldsymbol{F}$ and minimum cut $\boldsymbol{c}$. Lastly, capacities are sampled as integers

Table 1: Mean Absolute Error (MAE) and accuracy of predicting the final $\boldsymbol{F}$ and intermediate flow $\bar{\boldsymbol{F}}^{(t)}$, and min-cut $\boldsymbol{c}$ (if applicable) on *2-community* and *bipartite* graphs. *(primal)* corresponds to training on max-flow only. *(dual)* corresponds to training with both primal-dual heads. *(pipeline)* corresponds to learning min-cut first. *(no-algo)* corresponds to optimising directly max-flow, without learning Ford-Fulkerson.

| | 2-Community *(out-of-distribution)* | | | Bipartite *(out-of-family)* | | |
|---|---|---|---|---|---|---|
| **Model** | $\boldsymbol{F}$ | $\bar{\boldsymbol{F}}^{(t)}$ | $\boldsymbol{c}$ | $\boldsymbol{F}$ | $\bar{\boldsymbol{F}}^{(t)}$ | $\boldsymbol{c}$ |
| PGN-max *(primal)* | $0.266_{\pm 0.001}$ | $0.294_{\pm 0.002}$ | - | $0.56_{\pm 0.23}$ | $0.82_{\pm 0.17}$ | - |
| PGN-mean *(primal)* | $0.274_{\pm 0.001}$ | $0.311_{\pm 0.004}$ | - | $1.09_{\pm 0.47}$ | $1.13_{\pm 0.18}$ | - |
| MPNN-max *(primal)* | $0.263_{\pm 0.008}$ | $0.289_{\pm 0.004}$ | - | $0.75_{\pm 0.47}$ | $0.78_{\pm 0.11}$ | - |
| MPNN-mean *(primal)* | $0.278_{\pm 0.008}$ | $0.313_{\pm 0.003}$ | - | $0.75_{\pm 0.47}$ | $0.92_{\pm 0.22}$ | - |
| PGN-max *(dual)* | $\mathbf{0.234_{\pm 0.002}}$ | $\mathbf{0.269_{\pm 0.001}}$ | $100\%_{\pm 0.0}$ | $0.49_{\pm 0.22}$ | $\mathbf{0.78_{\pm 0.29}}$ | $100\%_{\pm 0.0}$ |
| PGN-mean *(dual)* | $0.240_{\pm 0.004}$ | $0.285_{\pm 0.004}$ | $100\%_{\pm 0.0}$ | $1.10_{\pm 0.30}$ | $1.05_{\pm 0.12}$ | $99\%_{\pm 0.7}$ |
| MPNN-max *(dual)* | $0.236_{\pm 0.002}$ | $0.288_{\pm 0.005}$ | $100\%_{\pm 0.0}$ | $0.71_{\pm 0.32}$ | $0.98_{\pm 0.22}$ | $100\%_{\pm 0.0}$ |
| MPNN-mean *(dual)* | $0.258_{\pm 0.008}$ | $\mathbf{0.268_{\pm 0.002}}$ | $100\%_{\pm 0.0}$ | $0.81_{\pm 0.09}$ | $1.06_{\pm 0.35}$ | $100\%_{\pm 0.0}$ |
| PGN-max *(pipeline)* | $0.256_{\pm 0.001}$ | $0.293_{\pm 0.003}$ | $61\%_{\pm 0.1}$ | $\mathbf{0.45_{\pm 0.18}}$ | $\mathbf{0.77_{\pm 0.26}}$ | $95\%_{\pm 0.1}$ |
| PGN-mean *(pipeline)* | $0.244_{\pm 0.001}$ | $0.304_{\pm 0.001}$ | $100\%_{\pm 0.0}$ | $0.98_{\pm 0.44}$ | $1.03_{\pm 0.32}$ | $99\%_{\pm 0.8}$ |
| MPNN-max *(pipeline)* | $0.261_{\pm 0.002}$ | $0.312_{\pm 0.005}$ | $61\%_{\pm 0.3}$ | $\mathbf{0.47_{\pm 0.23}}$ | $0.95_{\pm 0.34}$ | $90\%_{\pm 1.1}$ |
| MPNN-mean *(pipeline)* | $0.255_{\pm 0.002}$ | $0.292_{\pm 0.002}$ | $100\%_{\pm 0.0}$ | $0.64_{\pm 0.35}$ | $0.92_{\pm 0.20}$ | $100\%_{\pm 0.0}$ |
| Random | $0.740_{\pm 0.002}$ | - | $50\%_{\pm 0.0}$ | $1.00_{\pm 0.00}$ | - | $50\%_{\pm 0.0}$ |
| PGN-max *(no-algo)* | $0.314_{\pm 0.013}$ | - | - | $0.78_{\pm 0.02}$ | - | - |

from $U(0, 10)$ and then rescaled via a min-max normalisation for *2-community* graphs, while they are sampled as either 0 or 1 for *bipartite* graphs.

**Ablation & neural architectures**   We performed an ablation study to assess the contribution from the dual, by training the same DAR architecture without the additional *min-cut head* (consequently the dual information does not flow back in $P_F$ in Figure 1). To deepen our analysis, we also consider a neural architecture where the minimum cut is learnt prior the Ford-Fulkerson algorithm. Specifically, we introduce a third processor that is trained solely on minimum cut, whose output is then used as an additional feature for the architecture presented in Figure 1. Furthermore, we compare two different types of processors: (i) a fully-connected Message-Passing Neural Network (MPNN) (Gilmer et al., 2017), which implements equation 3 and exchanges messages between all pairs of nodes; (ii) Pointer-Graph Network (PGN) (Velickovic et al., 2020a), which instead exchanges messages only between a node and its neighbours defined by the *inputs* and *hints* of the algorithm. For all processors, we try different aggregation operators in equation 3, namely $\bigoplus = \{\mathrm{max}, \mathrm{mean}, \mathrm{sum}\}$. We train all models for 20,000 epochs with the SGD optimiser and we average the results across 5 runs. We also use *teacher forcing* with a decaying factor of 0.999. This has the effect of providing the network with ground-truth *hints* for the early stage of the training phase, while letting the network predictions flow in for the majority of training. To choose optimal hyperparameters, e.g. learning rate, hidden dimension, we employ a bi-level random search scheme, where the first level samples values of hyperparameters in a large range of values, while the second one "refines" the search based on the first level results. We choose the best hyperparameters based on the validation error on $\boldsymbol{F}$. Aggregated validation loss curves are shown in Figure 2(a). For further details on the model selection, refer to the appendix.

**Results analysis**   We report results on Ford-Fulkerson simulation in Table 1. Specifically, we use the Mean Absolute Error (MAE) as a metric for assessing the predictions of the final flow assignment $\boldsymbol{F}$, obtained as in equation 5. Similarly, we measure average performance on all the intermediate flow assignment $\bar{\boldsymbol{F}}^{(t)}$ in order to show how well the algorithm is imitated across all steps, which is referred to as $\bar{\boldsymbol{F}}^{(t)}$ in Table 1. Where applicable, we report accuracy on the minimum cut as well, i.e. for *dual* and *pipeline* models. To better evaluate all models, we include a random baseline which samples $\boldsymbol{F}$ at random and rescales it following equation 5 and a GNN trained to directly output the flow matrix $\boldsymbol{F}$ without learning Ford-Fulkerson (marked as *no-algo*). First, Table 1 shows clear performance advantage with respect to the two baselines, indicating that learning *max-flow* with the support of algorithmic reasoning, i.e. learning of Ford-Fulkerson, is more effective. More importantly, we notice how models incorporating the prediction of the dual problem consistently outperform the *primal* baselines on both *2-community* and *bipartite* graphs. Dual architectures also

Table 2: Qualitative analysis on the prediction of $\boldsymbol{F}$. Mean Absolute Error (MAE) is used as the regression error from the ground truth maximum flow value. For simplicity, we only report results of the best-performing models (PGNs).

| Metric | primal | | dual | | pipeline | |
|---|---|---|---|---|---|---|
| | PGN-max | PGN-mean | PGN-max | PGN-mean | PGN-max | PGN-mean |
| $\|\boldsymbol{F} - \boldsymbol{F}^*\|$ | $7.86 \pm 0.47$ | $8.68 \pm 0.21$ | $\mathbf{0.34 \pm 0.04}$ | $0.41 \pm 0.01$ | $7.58 \pm 0.10$ | $0.38 \pm 0.08$ |

better imitate the algorithm across all intermediate steps compared to primal, as testified by lower $\bar{\boldsymbol{F}}^{(t)}$. This suggests that the dual min-cut information, despite being easy to learn (Fereydounian et al., 2022), helps the model achieve a lower prediction error. This finding is also strengthened by the observation that whenever the min-cut prediction is imprecise, e.g. {PGN, MPNN}-max pipeline for *2-community*, the prediction of $\boldsymbol{F}$ and $\boldsymbol{F}^{(t)}$ become consequently worse. From our experiments, the *dual* PGN architecture with max aggregator emerges as the best-performing model, at least for what concerns *2-community* graphs, being able to perfectly predict also the minimum cuts of all the graphs in the test set. Contrastingly, learning min-cut first is less stable (while still outperforming the primal baseline) confirming prior work findings on the effectiveness of multi-task learning.

The performance gap also increases when testing *out-of-family* on bipartite graphs, where *dual* and *pipeline* with max aggregator are both very competitive. We note that for bipartite graphs we record higher mean and standard deviations. While this behaviour is emphasised by the fact that capacities are sampled as either 0 or 1, i.e. leaving more chances for prediction errors, this testifies that generalisation to arbitrary graph distributions is still a challenging task.

**Qualitative analysis**   To further evaluate the performance of DAR, we perform a qualitative study, whose results are presented in Table 2. For *2-community* graphs we assess how close the predicted flow matrix $\boldsymbol{F}$ is to the optimal max-flow solution without considering errors for intermediate nodes. This gives a measure of how well the network can predict the maximal flow value in the graphs and use it in the predicted solution. To achieve that, we ignore intermediate errors and only measures flow signal exiting the source node $s$ and entering the sink node $t$, i.e. $\sum_{(s,j) \in E} \boldsymbol{F}_{sj}$ and $\sum_{(j,t) \in E} \boldsymbol{F}_{jt}$. Thus, we take the maximum (absolute value) between the two and compare this value to the ground truth maximum flow value $\boldsymbol{F}^*$. From Table 2 we observe that all the dual architectures exhibit a solution which reflects the true maximum flow quantity in the input graphs, i.e. $\approx 0.30$ of MAE from the optimum on average. This analysis further solidifies our claim that a DAR model can positively transfer knowledge from the dual to the primal problem resulting in more accurate and qualitatively superior solutions. This claim is also supported by the fact that both primal architectures and dual architectures for which min-cut results are worse miss the optimal solution by a large margin (compare PGN-max *pipeline* min-cut results in Table 1 and higher MAE in Table 2).

## 4.2   REAL-WORLD GRAPHS

**Benchmark description**   We assess generality and potential impact of the DAR pipeline by considering a real-world edge classification task, for which prior knowledge of the concept of *max-flow* might be helpful. We test both the primal and the DAR architectures on the Brain Vessel Graphs (BVG) benchmark (Paetzold et al., 2021). This benchmark contains 9 large-scale real-world graphs, where edges represent vessels and nodes represent bifurcation of the vessel branches in a brain network. The task is to classify each edge in three categories: *capillaries*, *veins* and *arteries* based on the following features: vessel length $l_{ij}$; shortest distance between bifurcation points $d_{ij}$; and curvature $\rho_{ij}$. Note that the three classes can be distinguished by the radius of the vessel, or equivalently, by the amount of blood flow that can traverse the vessel. Hence, being able to simulate the blood flow in the entire brain network is likely to be advantageous to effectively solve the task. As an additional challenge, note that the classification task is highly imbalanced, i.e. 95% of samples are capillaries, 4% veins and only 1% arteries.

We test the models on three BVG graphs, namely CD1-E-1 (the largest, with 5,791,309 edges), CD1-E-2 (2,150,326 edges) and CD1-E-3 (3,130,650 edges). BVG data also include a synthetic brain vessel graph for validation purposes, comprising 3159 nodes and 3234 edges.

**Algorithm reconstruction**    The main difference with our synthetic tasks is that here we need to estimate the vessel diameter/radius which is a quantity that can be related with the vessel *capacity*, i.e. how much blood (flow) can traverse the edge. Therefore, the *capacity* is a learning target rather than one of the features to feed our algorithmic reasoner with. Here, we exploit the generality of the encode-process-decode architecture and learn to *reconstruct* the Ford-Fulkerson neural execution. Specifically, we reuse PGN-max networks pre-trained on *2-community* graphs (section 4.1).

As the *capacity* is no longer an input feature, we drop the *capacity* encoder from $f_v$ and introduce three new encoder layers in $f_v$, one for each feature of the vessel graph benchmark, i.e. $l_{ij}, d_{ij}, \rho_{ij}$. Thus, we freeze all the parameters in the pre-trained models apart from the introduced encoder layers. Hence, we only train the weights of $l_{ij}, d_{ij}, \rho_{ij}$ to learn Ford-Fulkerson steps in absence of input information about capacity. In other words, the model learns to use $l_{ij}, d_{ij}, \rho_{ij}$ to estimate the edge flows in the network, which act as proxy information for edge capacities, i.e. our primary objective in the BVG task. We perform these learning steps of algorithm reconstruction on the synthetic vessel graph provided by the BVG benchmark. Source and sink nodes $s, t$ are chosen as two random nodes whose shortest distance is equal to the diameter of the graph. We train to reconstruct the algorithm for 15000 epochs, with Adam optimiser (Kingma & Ba, 2015) and learning rate 1e-5. Figure 2(b) compares the loss curves for the primal and DAR models, on the task.

Thus, we simulate one single step of Ford-Fulkerson on CD1-E-*X* through PGN-max *primal* and *dual* models and extract hidden learnt representations for each node, which are then summed together to get edge embeddings. These edge embeddings will be used as additional input features for the graph neural networks (described below) which we train to solve brain vessel classification. Finally, we highlight how this approach allows us to easily dump the embeddings, as the reconstructed encoders and processors will not be training further on real-data.

**Neural architectures**    We consider graph neural networks from the BVG benchmark paper as our baselines, namely Graph Convolutional Networks (GCNs) (Kipf & Welling, 2017), GraphSAGE (Hamilton et al., 2017) and ClusterGCN (Chiang et al., 2019) with GraphSAGE convolution (C–SAGE). The general architecture consists of several graph convolutional layers with ReLU activations followed by a linear module. Additionally, we use the embeddings extracted by PGN-max *(primal)* and PGN-max *(dual)* to train a simple linear classifier (LC) to assess how much information these embedding add with respect to the original $l_{ij}, d_{ij}, \rho_{ij}$ features. We also use those representations in combination with GraphSAGE and C–SAGE. Specifically, our embeddings are concatenated together with the GraphSAGE's and C–SAGE's learnt embeddings prior the final linear layer. As an additional sanity check, we also train Node2Vec (Grover & Leskovec, 2016) on each of the three datasets and concatenate its learnt embeddings the same way. All models are trained with early stopping of 300 epochs and optimal hyperparameters taken from the BVG paper, which we report in the appendix for completeness. Finally, we average the results across 3 trials.

**Results analysis**    Results on the BVG benchmark are reported in Table 3. As the learning problem is highly imbalanced, we use the balanced accuracy score (average of recall for each class) and the area under the ROC curve as metrics to evaluate the performance.

Looking at LC performance, we see that the algorithmic reasoner embeddings (both primal and dual) are informative, resulting in an average 16.6% increase in balanced accuracy and 10.5% in ROC across the three datasets when compared to simple features. Dual embeddings also show superior performance compared to primal embeddings, as testified by consistent increments in both metrics. Figure 2(b) hints that this might be due to a better algorithm reconstruction in the dual, which results in more informative representations. LC performance also gives a clear indication of how well the algorithmic reasoner is able to positively transfer knowledge acquired on synthetic algorithmic tasks to unseen real-world predictive graph learning ones.

When considering the use of learnt embedding in combination with GNN architecture, we note significant performance improvements over vanilla (i.e. non algorithmically enhanced) GNNs. C–SAGE with *dual* embeddings achieves the best performance on all three datasets with a consistent performance gap for CD1-E-3 and CD1-E-2. Interestingly, dual embeddings consistently outperform Node2Vec embeddings. This is remarkable, considering that Node2Vec is trained directly on the CD1-E-*X* data, whereas DAR only performs inference on them. A reason to this performance gap might be that Node2Vec essentially widens the local perceptive field of graph neural networks

Table 3: Balanced accuracy (Bal. Acc.) and area under the ROC curve (ROC) performance metrics on large-scale brain vessel graphs. LC refers to a linear classifier. In addition to the standard architectures, we consider variants where the final linear classification layer takes in additional Node2Vec (Grover & Leskovec, 2016) learnt embeddings and embeddings extracted from PGN-max primal and dual architectures.

| Model | CD1-E-3 | | CD1-E-2 | | CD1-E-1 | |
|---|---|---|---|---|---|---|
| | Bal. Acc. | ROC | Bal. Acc. | ROC | Bal. Acc. | ROC |
| LC | $39.3\%_{\pm 0.2}$ | $52.3\%_{\pm 0.6}$ | $36.9\%_{\pm 0.5}$ | $55.9\%_{\pm 0.1}$ | $45.5\%_{\pm 0.1}$ | $61.7\%_{\pm 0.0}$ |
| LC *(N2V)* | $43.9\%_{\pm 0.2}$ | $55.5\%_{\pm 0.1}$ | $71.9\%_{\pm 0.1}$ | $62.6\%_{\pm 0.0}$ | $46.1\%_{\pm 0.1}$ | $60.0\%_{\pm 0.0}$ |
| LC *(primal)* | $48.6\%_{\pm 0.4}$ | $59.4\%_{\pm 0.4}$ | $58.7\%_{\pm 0.1}$ | $63.8\%_{\pm 0.2}$ | $45.3\%_{\pm 0.1}$ | $59.9\%_{\pm 0.1}$ |
| LC *(dual)* | $53.8\%_{\pm 0.3}$ | $66.2\%_{\pm 0.2}$ | $67.3\%_{\pm 0.1}$ | $71.8\%_{\pm 0.0}$ | $48.1\%_{\pm 0.5}$ | $62.1\%_{\pm 0.3}$ |
| GCN | $58.1\%_{\pm 0.5}$ | $67.9\%_{\pm 0.2}$ | $74.6\%_{\pm 1.7}$ | $78.7\%_{\pm 0.1}$ | $59.0\%_{\pm 0.2}$ | $67.9\%_{\pm 0.2}$ |
| SAGE | $63.5\%_{\pm 0.2}$ | $70.9\%_{\pm 0.3}$ | $73.9\%_{\pm 0.6}$ | $82.5\%_{\pm 0.2}$ | $64.7\%_{\pm 0.7}$ | $74.2\%_{\pm 0.2}$ |
| SAGE *(N2V)* | $65.0\%_{\pm 0.1}$ | $71.9\%_{\pm 0.1}$ | $84.1\%_{\pm 1.9}$ | $82.5\%_{\pm 0.4}$ | $65.9\%_{\pm 0.1}$ | $74.8\%_{\pm 0.1}$ |
| SAGE *(primal)* | $64.5\%_{\pm 0.2}$ | $72.0\%_{\pm 0.2}$ | $83.8\%_{\pm 0.4}$ | $83.7\%_{\pm 0.4}$ | $66.2\%_{\pm 0.5}$ | $74.8\%_{\pm 0.3}$ |
| SAGE *(dual)* | $66.7\%_{\pm 0.4}$ | $75.0\%_{\pm 0.2}$ | $85.2\%_{\pm 0.1}$ | $85.5\%_{\pm 0.2}$ | $66.4\%_{\pm 0.3}$ | $74.8\%_{\pm 0.1}$ |
| C–SAGE | $68.6\%_{\pm 0.8}$ | $74.2\%_{\pm 0.5}$ | $81.8\%_{\pm 0.5}$ | $85.6\%_{\pm 0.2}$ | $59.3\%_{\pm 0.9}$ | $68.3\%_{\pm 0.5}$ |
| C–SAGE *(N2V)* | $68.6\%_{\pm 0.2}$ | $74.1\%_{\pm 0.1}$ | $84.8\%_{\pm 0.2}$ | $84.8\%_{\pm 0.5}$ | $67.4\%_{\pm 0.6}$ | $\mathbf{75.9\%_{\pm 0.2}}$ |
| C–SAGE *(primal)* | $67.3\%_{\pm 0.2}$ | $73.6\%_{\pm 1.9}$ | $82.5\%_{\pm 1.9}$ | $84.0\%_{\pm 1.5}$ | $67.7\%_{\pm 0.1}$ | $75.8\%_{\pm 0.2}$ |
| C–SAGE *(dual)* | $\mathbf{70.2\%_{\pm 0.2}}$ | $\mathbf{76.3\%_{\pm 0.1}}$ | $\mathbf{85.6\%_{\pm 0.2}}$ | $\mathbf{86.7\%_{\pm 0.3}}$ | $\mathbf{68.1\%_{\pm 0.2}}$ | $75.8\%_{\pm 0.1}$ |

with random walks as an attempt to capture global graph features. On the contrary, DAR utilises a more principled approach based on the simulation of graph flow. This means that the learnt latent space encodes the information necessary to reconstruct the flow assignment and consequently edge capacities, these being more informative for the specific task. DAR models also exhibit very good generalisation capabilities. In fact, we recall that the networks are only trained on graphs with 16 nodes and extract meaningful representations for graphs with millions of nodes, being able to provide a clear performance advantage over baselines. This might also indicate a way worth pursuing to realise sustainable *data-efficient* learning models for graphs.

## 5 CONCLUSION

We have presented **dual algorithmic reasoning** (DAR), a neural algorithmic reasoning approach that leverages duality information when learning classical algorithms. Unlike other approaches, we relax the assumption of having multiple algorithms to be learnt jointly and show that incorporating the dual of the problem targeted by algorithms represents a valuable source of information for learnt algorithmic reasoners. We showed that learning together the primal-dual max-flow-min-cut problem can substantially improve the quality of the predictions, as testified by the quantitative and qualitative evaluations of the models. Furthermore, dual algorithmic reasoners have demonstrated to generalise better, showing positive knowledge transfer across different families of graph distributions and extracting informative representations for large-scale graphs while only being trained on toy-synthetic graphs. In this context, we also demonstrated for the first time how more classical graph learning tasks can be tackled through exploitation of algorithmic reasoning, via *algorithm reconstruction*. On a final note, we identify several problems and algorithms that may benefit from a dual reasoning approach. First, max-flow and min-cut may be representative for a wide class of primal/dual pairs, for which *strong duality* holds. There, the dual solution can be used to recover the primal optimum (and vice versa), equivalently to max-flow and min-cut. Examples of such problems are *shortest path* and *min-cost flow* problems. More interestingly, we may generalise this approach to target also *weak* primal-dual problems, in which the dual objective is an approximation of the primal objective. Even in the case of weak duality, dual information is valuable, as testified by the numerous algorithms exploiting primal-dual relations (Balinski, 1986; Pourhassan et al., 2017). Particularly interesting problems to target may be the *Travelling Salesman Problem* (Cormen et al., 2009), for which a dual formulation includes learning to output a 1-tree (Bazaraa & Goode, 1977), and the *Weighted Vertex Cover* problem, for which the dual can effectively be used to develop 2-approximation heuristics (Pourhassan et al., 2017). We believe that results showed in this manuscript should strongly motivate further work in this direction, extending the analysis to other pairs of primal-dual problems, such as the ones suggested in this section.

ACKNOWLEDGMENTS

This research was partially supported by TAILOR, a project funded by EU Horizon 2020 research and innovation programme under GA No 952215.

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

# A PSEUDO-CODE

## A.1 FORD-FULKERSON PSEUDO-CODE

We consider a standard Ford-Fulkerson implementation, shown in Algorithm 1.

---
**Algorithm 1** Ford-Fulkerson
---
**Input:** $G = (V, E, c)$, $s$, $t$;
$f_{uv} = 0 : \forall (u, v) \in E$
**while** $\exists$ augmenting path $p = s, \ldots, t$ in $G_f$ **do**;
    $df = \min\{c_{uv} : (u, v) \in p\}$;
    **for** each $(u, v) \in p$ **do**
        $f_{uv} = f_{uv} - df$
        $f_{vu} = f_{vu} + df$
    **end for**
**end while**
**return** $f$

---

## A.2 CORRECTIVE FLOW PROCEDURE PSEUDO-CODE

The corrective procedure is shown in Algorithm 2. We discriminate between negative nodes $i^-$, if $(\sum_{(i,j)\in E} F_{ij} + \sum_{(j,i)\in E} F_{ji}) < 0$, i.e. the node keeps some of the flow, and positive nodes $i^+$ if the sum of in-flow and out-flow is instead positive. Thus, we enforce the flow conservation constraint in two simple steps. First, all negative nodes $i^-$ sends back to the source $s$ an amount of flow equal to the magnitude of the constraint violation without considering edge-capacities. All positive nodes instead, lower the amount of out-flow towards the sink $t$. After this step, the conservation of flows is satisfied but we have introduced capacity violations in the graph. Hence, in the second step we impose back capacity constraints. Starting from the source $s$, we find paths with capacity violations and clamp the flow value to the maximum capacity. Thus, we readjust the flow sent from $s$ until all capacities violations are corrected and no flow conservation error occurs.

---
**Algorithm 2** Flow correction algorithm
---
*Input*: $G = (V, E)$, $F$: flow matrix, $C$: capacity matrix, $s$: source, $t$: sink
$V^- = \{v \mid v \in V \wedge \sum_{(i,v)\in E} F_{iv} + \sum_{(v,i)\in E} F_{vi} < 0\}$
$V^+ = \{v \mid v \in V \wedge \sum_{(i,v)\in E} F_{iv} + \sum_{(v,i)\in E} F_{vi} > 0\}$
**for** each $v- \in V^-$ **do**
    $\varepsilon = |\sum_{(i,v)\in E} F_{iv} + \sum_{(v,i)\in E} F_{vi}|$
    find a path from $v^-$ to $s$ and send back $\varepsilon$ amount of flow to $s$
**end for**
**for** each $v+ \in V^+$ **do**
    $\varepsilon = |\sum_{(i,v)\in E} F_{iv} + \sum_{(v,i)\in E} F_{vi}|$
    find a path from $v^+$ to $t$ and reduce the out-flow by $\varepsilon$
**end for**
$\varepsilon_s = |\sum_{(s,i)\in E} F_{si} - \sum_{(s,i)\in E} C_{si}|$
**while** $\varepsilon_s > 0$ **do**
    find a path from $s$ to $t$ and reduce the out-flow by enforcing capacity constraints
    recompute $\varepsilon_s$
**end while**
**return** $F$

---

Table 4: First-level and second-level random searches details.

| Level | Hyperparameter | Min. value | Max. value | Distribution |
|-------|----------------|------------|------------|--------------|
| Level 1 | Learning rate | 1e-5 | 1e-1 | log-uniform |
| | Weight decay | 1e-5 | 1e-1 | log-uniform |
| | Hidden dimension | 16 | 512 | uniform |
| Level 2 | Learning rate | 1e-3 | 1e-2 | log-uniform |
| | Weight decay | 1e-3 | 4e-3 | log-uniform |
| | Hidden dimension | 60 | 100 | uniform |

## B  HYPERPARAMETER OPTIMISATION

### B.1  SYNTHETIC GRAPHS

We sample $n = 50$ configurations for each level of the bi-level random search. In Table 4 we report details for the first and second level random searches, where the second refines the first based on validation results of the primal model. Winner hyperparameters for *primal* model are: hidden dimension = 68, learning rate = 0.009341, weight decay = 0.003420. Winner hyperparameters for *dual* model are: hidden dimension = 65, learning rate = 0.009868, weight decay = 0.001734. *Pipeline* shares hyperpameters with *dual*.

### B.2  BRAIN VESSEL BENCHMARK

Optimal hyperparameters for BVG data are presented in Table 5. We perform minimal hyperparameters optimisation for C–SAGE for CD1-E-1, as the optimal hyperparameters reported in Paetzold et al. (2021) did not perform well. Specifically, we optimised *number of layers* and *number of hiddens* by grid-searching over $\{2, 3, 4\}$ and $\{64, 128\}$, respectively.

Table 5: Optimal hyperparameters used for the BVG benchmark. Note that C–SAGE needs more epochs for CD1-E-3.

| Dataset | GCN | SAGE | C–SAGE | Node2Vec |
|---------|-----|------|--------|----------|
| CD1-E-3 | lr=$3 \cdot 10^{-3}$
no. layers=3
no. hiddens=256
dropout=0.4
epochs=1500 | lr=$3 \cdot 10^{-3}$
no. layers=4
no. hiddens=128
dropout=0.4
epochs=1500 | lr=$3 \cdot 10^{-3}$
no. layers=4
no. hiddens=128
dropout=0.2
epochs=5000 | lr=$1 \cdot 10^{-2}$
walk length=40
walks per node=10
no. hiddens=128
epochs=5 |
| CD1-E-2 | *as above* | *as above* | lr=$3 \cdot 10^{-3}$
no. layers=4
no. hiddens=128
dropout=0.2
epochs=1500 | *as above* |
| CD1-E-1 | *as above* | *as above* | lr=$3 \cdot 10^{-3}$
no. layers=3
no. hiddens=64
dropout=0.2
epochs=1500 | *as above* |

## C  ADDITIONAL RESULTS

**Qualitative study**   Here, we present an additional qualitative study for all models in Table 1 on synthetic graphs, both *2-community* and *bipartite*. To further strengthen qualitative findings of Table 2, we aim to measure to what extent the *optimal* max-flow value can be linearly decodable from

Table 6: $R^2$ score of predicting the maximum flow value from learnt graph representations $\boldsymbol{h}_g$. $R^2$ metric emits score in $(-\infty, 1]$, with 1 being the best possible score.

| | primal | | dual | | pipeline | |
|---|---|---|---|---|---|---|
| Graphs | PGN-max | PGN-mean | PGN-max | PGN-mean | PGN-max | PGN-mean |
| *2-community* | $0.69 \pm 0.12$ | $-0.59 \pm 0.33$ | $\mathbf{1.00 \pm 0.00}$ | $0.93 \pm 0.01$ | $0.80 \pm 0.05$ | $-0.13 \pm 0.29$ |
| *bipartite* | $-0.79 \pm 0.81$ | $-115.6 \pm 64$ | $\mathbf{0.98 \pm 0.01}$ | $-115.2 \pm 64$ | $-0.69 \pm 0.33$ | $\mathbf{0.98 \pm 0.01}$ |

the learnt node representations. Specifically, we compute ground truth optimal max-flow value $f_g^*$ for all graphs in the test sets. Thus, for all models we neurally execute Ford-Fulkerson, collect learnt node embeddings $\boldsymbol{h}_i$ and obtain graph representations $\boldsymbol{h}_g$ through a MAX-POOL operation. Thus, we fit linear models on $\{\boldsymbol{h}_g\}$ and compute the $R^2$ score to evaluate the quality of the predictions. Table 6 shows that from *dual* embeddings we are able to linearly decode the optimal *max-flow* value almost perfectly, i.e. $\geq 0.99$ on average across the two graph distributions, confirming once again that learning min-cut and max-flow together is beneficial and yields more informative representations.

**Enforcing flow conservation** As explained in section 3.2 of the main paper, the max-flow output of the algorithmic reasoner (equation 5) is a noisy version ($\tilde{\boldsymbol{F}}$) of a real correct solution $\boldsymbol{F}$, as it may not satisfy the conservation of flows constraint (equation 2). We can enforce the flow conservation through Algorithm 2. Figure 3 provides a visual example of how the flow conservation is corrected by the corrective procedure of Algorithm 2, showing that it is possible to extract a correct solution from the neural output. In the figure, we represent negative nodes $i^-$ (see subsection A.2) with blue and positive nodes $i^+$ with red. Sources $s$ should always be "positive" nodes (flow is always sent out) while sinks should be "negative" (flow is always received). White nodes in Figure 3(b) represent nodes for which equation 2 holds.

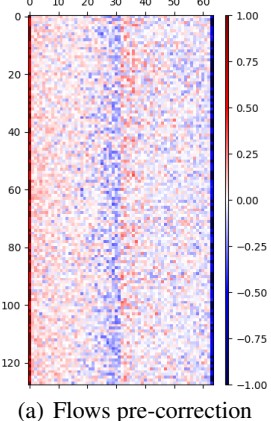

(a) Flows pre-correction

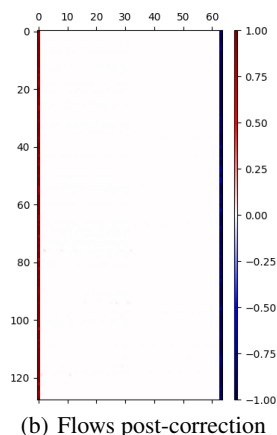

(b) Flows post-correction

Figure 3: This figure shows the (normalised) flow conservation error for all nodes (x axis) of each test graph (y axis). (a) shows the predicted $\boldsymbol{F}$ conservation error prior Algorithm 2. (b) shows the corrected $\boldsymbol{F}$ conservation error. Through Algorithm 2 we are able to correct all the inner error (white represents zero error), with $s$ and $t$ being the only nodes which respectively send and receive flow.

**Putting together Node2Vec and dual embeddings** In Table 3, Node2Vec and dual embeddings arose as the most informative node embeddings for solving the brain vessel classification task. Therefore, we run an additional experiment by concatenating together these two embeddings and learning neural networks to solve the BVG task the same way as it was done in subsection 4.2 for primal/dual/N2V embeddings separately. We report these results in Table 7. We record higher results for these combined embeddings compared to using the two embeddings separately, hinting that Node2Vec and dual representations encode two different kinds of information that might be effectively combined, i.e. structural (the former) and blood flow (the latter).

Table 7: Additional BVG results for concatenation of Node2Vec and dual embeddings. To be compared to Table 3 in the main paper.

| Model | CD1-E-3 | | CD1-E-2 | | CD1-E-1 | |
|---|---|---|---|---|---|---|
| | Bal. Acc. | ROC | Bal. Acc. | ROC | Bal. Acc. | ROC |
| LC *(dual-N2V)* | $53.9\%_{\pm0.1}$ | $66.3\%_{\pm0.1}$ | $77.8\%_{\pm0.1}$ | $74.3\%_{\pm0.1}$ | $48.2\%_{\pm0.2}$ | $62.1\%_{\pm0.1}$ |
| SAGE *(dual-N2V)* | $70.1\%_{\pm0.3}$ | $76.5\%_{\pm0.1}$ | $89.2\%_{\pm0.2}$ | $86.3\%_{\pm0.2}$ | $67.5\%_{\pm0.2}$ | $76.0\%_{\pm0.1}$ |
| C–SAGE *(dual-N2V)* | $70.4\%_{\pm0.1}$ | $76.7\%_{\pm0.1}$ | $88.3\%_{\pm0.3}$ | $85.0\%_{\pm0.3}$ | $67.0\%_{\pm0.2}$ | $75.8\%_{\pm0.3}$ |

Table 8: Validation results on *2-community* graphs for all models.

| Model | $F$ | $\bar{F}^{(t)}$ | $c$ |
|---|---|---|---|
| PGN-max *(primal)* | $0.218 \pm 0.002$ | $0.272 \pm 0.006$ | - |
| PGN-mean *(primal)* | $0.199 \pm 0.001$ | $0.281 \pm 0.004$ | - |
| MPNN-max *(primal)* | $0.215 \pm 0.003$ | $0.271 \pm 0.003$ | - |
| MPNN-mean *(primal)* | $0.235 \pm 0.008$ | $0.304 \pm 0.005$ | - |
| PGN-max *(dual)* | $\mathbf{0.183 \pm 0.001}$ | $0.250 \pm 0.002$ | $100\% \pm 0.0$ |
| PGN-mean *(dual)* | $\mathbf{0.184 \pm 0.001}$ | $0.266 \pm 0.004$ | $100\% \pm 0.0$ |
| MPNN-max *(dual)* | $0.185 \pm 0.001$ | $\mathbf{0.246 \pm 0.002}$ | $100\% \pm 0.0$ |
| MPNN-mean *(dual)* | $\mathbf{0.184 \pm 0.001}$ | $\mathbf{0.246 \pm 0.002}$ | $100\% \pm 0.0$ |
| PGN-max *(pipeline)* | $0.189 \pm 0.002$ | $0.262 \pm 0.003$ | $98.2\% \pm 0.1$ |
| PGN-mean *(pipeline)* | $0.193 \pm 0.001$ | $0.264 \pm 0.002$ | $100\% \pm 0.0$ |
| MPNN-max *(pipeline)* | $0.188 \pm 0.003$ | $0.257 \pm 0.003$ | $98.3\% \pm 0.1$ |
| MPNN-mean *(pipeline)* | $0.204 \pm 0.001$ | $0.278 \pm 0.002$ | $100\% \pm 0.0$ |
| Random | $0.553 \pm 0.003$ | - | $50\% \pm 0.0$ |
| PGN-max *(no-algo)* | $0.256 \pm 0.002$ | - | - |

## D  VALIDATION RESULTS

To better evaluate the algorithmic generalisation of our models, we report validation results on synthetic data in Table 8. Additionally, we report additional plots showing loss of MPNN-* models in Figure 4, similarly to Figure 2(a) for PGNs. Lastly, we also report validation performance for all variants of C–SAGE models, i.e. *vanilla*, *primal*, *dual* and *pipeline*, on brain vessel classification (see Figure 5).

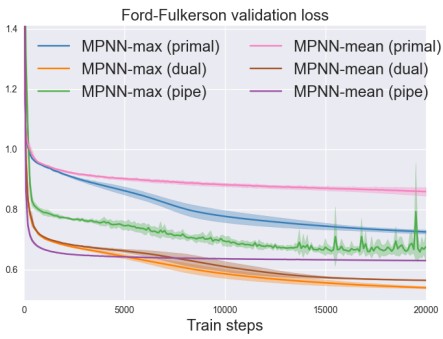

Figure 4: Ford-Fulkerson validation loss for MPNN processors.

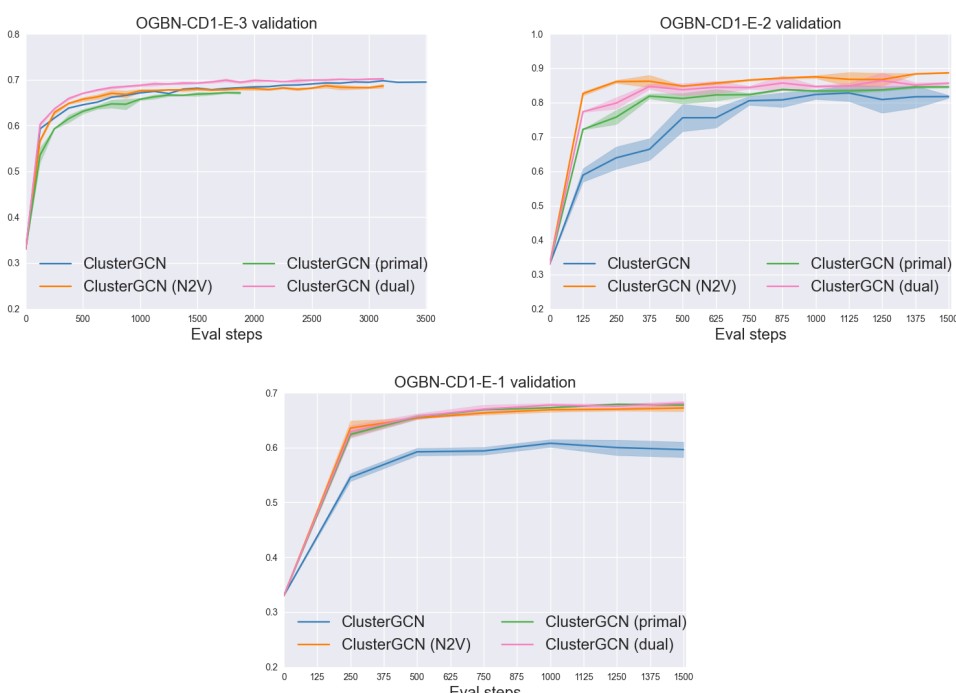

Figure 5: Validation performance of C–SAGE variants on CD1-E-*X* in terms of balanced accuracy. Truncated lines in CD1-E-3 plot represent early stopping.

