# OpenReview forum: "Dual Algorithmic Reasoning"
_ICLR.cc/2023/Conference — ICLR 2023 notable top 25%_

### Official Review · Reviewer_4SGE · 2022-10-21

**Confidence:** 4
**Correctness:** 3
**Technical Novelty And Significance:** 3
**Empirical Novelty And Significance:** 3
**Recommendation:** 8

**Clarity, Quality, Novelty And Reproducibility:**

The writing is clear and the quality is good. The paper presents a novel approach. No code is provide, but the details in the manuscript seem thorough.

**Strength And Weaknesses:**

Strengths:
  - Interesting and novel approach to leaning algorithmic reasoning using primal/dual.
  - Timely: the algorithmic reasoning community is growing.
  - Clear writing and good exposition.

Weaknesses:
  - There is only one algorithmic problem considered. In related work, more than one algorithm is used to show performance. This is a major weakness of the paper since the main claims are not entirely supported. The abstract states, "We demonstrate that simultaneously learning the dual definition of these optimisation problems in algorithmic learning allows for better learning and qualitatively better solutions." and In the conclusion, the authors write:  "We showed that learning together the primal-dual problem can substantially
improve the quality of the predictions, as testified by the quantitative and qualitative evaluations
of the models. Furthermore, dual algorithmic reasoners have demonstrated to generalise better,
showing positive knowledge transfer across different families of graph distributions and extracting
informative representations for large-scale graphs while only being trained on toy-synthetic graphs." But I find the claims much more general than the empirical findings which relate to only one problem on graph data and only two different families of graphs.

Minor issues not affecting score:
    - The citations and figure references are not linked.

___


After reading the author response, I have changed my score. The authors have addressed my concerns.


**Summary Of The Paper:**

This paper proposes a novel method for learning algorithms with neural networks. Specifically, the authors design a new architecture for processing graph data and a new training routine that considers both primal and dual formulations of optimization problems. Using min cut max flow problems, they compare their method to existing neural algorithmic reasoning tools. They show that their method beats the existing methods.


**Summary Of The Review:**

The technique is interesting and novel, but the empirical results are lacking.

---

> ### Author Response · Authors · 2022-11-14
> **Reply to Reviewer 4SGE**
>
> We thank the reviewer for taking the time to assess our contribution! The reviewer’s concerns about only testing on a single pair of primal-dual problems as well as phrasing through the paper are sensible, we thank the reviewer for pointing this out -- we have adjusted and made more specific the claims in the conclusions. However, we would like to elaborate further on our reasoning behind focusing on max-flow/min-cut.
>
> First, we believe that max-flow and min-cut might be representative for a wide class of optimisation problems as well, namely the ones for which _strong duality_ holds. In these kinds of problems, primal and dual share the same optimal value. In our experiments, we evaluated to what extent the neural network’s solutions reflect this property. In the qualitative study of Table 2, we show that clearly the dual reasoners output solutions that are very close to the maximum flow value. This suggests that the algorithmic reasoner can identify that both problems share a common solution, and can use that solution to one problem to help output better solutions for the other (if either of the two problems is learnt sufficiently well) -- a property that we expect to hold for all strong primal/duals. Furthermore, we believe that such property might still be relevant for solving optimisation problems for which the dual objective is an approximation of the primal objective, i.e. upper/lower bounds (weak duality). We have added a paragraph in the conclusions where we argue this a bit deeper.
>
> Second, as discussed in the paper, in order to ease OOD generalisation in algorithmic reasoning, we might want to have an _alignment_ between the architecture of the neural model and the dynamics of the target algorithm [1,2]. Therefore, testing other pairs of primal/dual problems (which would likely require quite different algorithms w.r.t Ford-Fulkerson) would require a bit of tweaking in the design choices of the neural architecture. Instead, we opted to show the better generalisation capabilities of dual reasoners on a real-world problem (which is strongly OOD), in which information about graph flow would be useful. There, the empirical results turned out to be consistently in favour of dual reasoners, both from a pure evaluation of performances on the edge classification task (see Table 3) and re-learning of the Ford-Fulkerson algorithm with different features, as reported in Figure 2(b).
>
> Regarding the reviewer’s statement that “empirical results are lacking”, we have added another experiment on real-world data, combining Node2Vec and dual embeddings (the best performing ones) showing that we can get even better results on BVG data (we report these results in the appendix). This also indicates that indeed the two embeddings carry different types of information: structural (N2V) and flow information (dual). This adds to the already shown results on synthetic data (Table 1) and real-world ones (Table 3) as well as qualitative analysis in Table 2 and in the appendix. We believe we presented a sufficient number of results.
>
> We are happy to discuss further on any of the raised points!
>
>
> [1] “How Neural Networks Extrapolate: From Feedforward to Graph Neural Networks”, Keyulu Xu, Mozhi Zhang, Jingling Li, Simon S. Du, Ken-ichi Kawarabayashi, Stefanie Jegelka (ICLR 2021)
>
> [2] “What can neural networks reason about?” Keyulu Xu, Jingling Li, Mozhi Zhang, Simon S. Du, Ken-ichi Kawarabayashi, Stefanie Jegelka (ICLR 2020)

---

> > ### Comment · Reviewer_4SGE · 2022-11-16
> > **Follow up**
> >
> > I appreciate the thorough response from the authors. In light of the changes, I am changing my score to reflect that I think this paper should be accepted.

---

### Official Review · Reviewer_vSCg · 2022-10-24

**Confidence:** 3
**Correctness:** 4
**Technical Novelty And Significance:** 4
**Empirical Novelty And Significance:** 4
**Recommendation:** 8

**Clarity, Quality, Novelty And Reproducibility:**

The paper has very high clarity, everything easily understandable and except for some remaining details (see weaknesses) well explained.
The quality of the paper is high, the method appears to be well thought-through, and the conducted experiments are well-designed and demonstrate the effectiveness of the method.
To the best of my knowledge the idea of using duality to enhance GNNs for algorithmic reasoning is novel.
Almost all parts of the method and the experiments are explained in sufficient detail. Code for reproducing the experimental results is not provided.

**Strength And Weaknesses:**

Overall, I liked this paper a lot. It is very well written and the idea of using duality information for learning to execute optimization algorithms with neural networks is exciting.
The empirical evaluation is also very strong in my opinion, clearly showing the benefit of the proposed method even when going beyond an evaluation on purely synthetic data.

One critique point is that the paper only covers one specific primal/dual pair of algorithms (max-flow/min-cut), and the architecture is heavily engineered towards this problem. It would be great to see how the presented ideas of using dual information as a subroutine of the primal algorithm translate to other primal/dual pairs of algorithms, a short exposition of potential candidates would be very useful.
Additionally I have some remaining questions on the details of the method and the experiments.
- What is the architecture used for the two decoders?
- Is there an interpretation for the big differences in performance from max and mean pooling? Could they be used in conjunction to improve performance or at least to remove the need to optimize over it?
- Regarding re-training for the real-world experiment: What exactly is the training setup for training the new encoder? Is the true radius of the vessel used to produce ground truth capacities that the target algorithm is executed on? Have the authors tried to directly predict the capacity from the three available features as a baseline?


Sidenote: In Figure 1 the second processor also accesses the inputs of the algorithm, but the arrows are not drawn.

**Summary Of The Paper:**

This paper follows a line of work on learning to mimic algorithms with neural networks to allow for better out-of-distribution generalization. Following recent advancement, the authors aim to train a GNN to simulate the intermediate steps of graph algorithms, using intermediate states of the true algorithm as supervision targets.

The authors expand on recent findings that training such an architecture benefits from multi-task learning, i.e. simultaneously learning to execute multiple related algorithms. They propose to leverage duality by simultaneously learning the primal and the dual version of an optimization algorithm. This gives the benefits of multi-task learning, reduces the chance for error propagation, and allows to leverage the learnt dual problem as a subroutine of the primal algorithm.
Specifically, this paper considers learning to execute the Ford-Fulkerson algorithm for the max-flow problem along with its dual min-cut problem.

The method follows the encode-process-decode framework, and consists of two iteratively executed GNN processors operating on the latent space. The first processor retrieves augmenting paths, while the second one learns to perform flow-update operations and predicts min-s-t-cuts. Some of the constraints on the max-flow are enforced by appropriate transformations, others are corrected for after termination of the neural algorithm.

The method is empirically tested on both synthetic and real-world data.
The neural algorithm is trained on small synthetic 2-community graphs.
In the synthetic experiment,  it is then evaluated on synthetic graphs that are out-of-distribution and out-of-family. The results show a clear benefit of incorporating the dual information both in terms of final performance and in terms of algorithmic alignment. They also demonstrate the benefit of using the dual solution as a subroutine of the primal neural algorithm, thereby extending beyond the setting of simple multi-task learning.

In the real-world experiment, the task is to classify the edges of a large-scale brain vessel graph from edge-level features. The goal is to improve existing classification pipelines by extracting additional features that capture the flow-capacity, using the neural algorithm that was trained on synthetic data. After retraining the encoder to adapt for the new inputs, the neural algorithm is executed for a single step on the large-scale graph, and the latent representation of the neural algorithm is aggregated to extract the edge features. Running existing classification pipelines with the additional features shows a clear performance improvement, demonstrating that the neural algorithm captures useful information even when executed on graphs much larger than the training data.

**Summary Of The Review:**

This paper proposes a novel idea, translates it into a well-designed method and empirically validates the effectiveness on both synthetic and real-world data. This is a strong paper in my opinion that is highly relevant for the ICLR community, and I therefore recommend acceptance.

---

> ### Author Response · Authors · 2022-11-14
> **Reply to Reviewer vSCg**
>
> We thank the reviewer for their careful evaluation and we are very pleased they liked the paper! We would like to answer each of the concerns raised here:
>
> 1) We have added a new paragraph in the conclusions where we elaborate more on which pairs of primal/duals are the most promising to tackle, thanks for the suggestion!
>
> 2) We apologise for the missing detail, the architecture used for the two decoders is the same as the encoders, i.e. linear layer.
>
> 3) Thanks for the interesting question. Several works in literature [1, 2, 3] draw a connection between the max pooling and the dynamics of classical graph algorithms. In particular, the work by Xu et al. [4] suggests that having an _alignment_ between the neural architecture and the algorithm we want to learn improves out-of-distribution generalisation and algorithmic learning in general. In this respect, max pooling typically aligns with a _discrete_ decision over neighbours (from the perspective of a single node), which encompasses a wide range of graph algorithms, e.g. Bellman-Ford / BFS / DFS and many more. Therefore, we believe that the performance gap (favourably to max) is due to the fact that Ford-Fulkerson needs to choose the bottleneck edge at every step, which is the _minimum_ capacity edge along the augmenting path from s to t. Therefore, there exists an alignment between the min operation inside Ford-Fulkerson and max pooling. Intuitively, we can think of it this way: in order to be able to “choose” a particular edge in the neural algorithm, we have to preserve its representation during message passing. By averaging node and edge embeddings (mean pooling), we likely lose that information after a few steps of message passing, whereas we can hope to preserve it for longer by using a max operation.
>
> 4) We apologise if this was not made clear in the paper. The reviewer's intuition is right, we used the true radius of the vessel as edge capacities to generate the trajectories of the algorithm. We recall that we only re-trained the network on a synthetic vessel graph, because we assumed that the true radiuses for the real-world graphs were unknown.
>
> 5) We tried to learn an edge-wise predictor from the three available features to the true radii in the synthetic graph. However, the learned predictor generally had weak generalisation performance. This implies that it is not trivial to extract the ground-truth radii from the given input features, and the network indeed needs to make use of the algorithmic computations in the processor network in order to get stronger representations for the downstream task.
>
> Thanks for pointing out the missing arrows in Figure 1, we updated the figure in the revised version!
>
> [1] “Neural execution of graph algorithms” Petar Veličković, Rex Ying, Matilde Padovano, Raia Hadsell, Charles Blundell (ICLR 2020)
>
> [2] “Combinatorial optimization and reasoning with graph neural networks” Quentin Cappart, Didier Chételat, Elias Khalil, Andrea Lodi, Christopher Morris, Petar Veličković (IJCAI 2021)
>
> [3] “How Neural Networks Extrapolate: From Feedforward to Graph Neural Networks”, Keyulu Xu, Mozhi Zhang, Jingling Li, Simon S. Du, Ken-ichi Kawarabayashi, Stefanie Jegelka (ICLR 2021)
>
> [4] “What can neural networks reason about?” Keyulu Xu, Jingling Li, Mozhi Zhang, Simon S. Du, Ken-ichi Kawarabayashi, Stefanie Jegelka (ICLR 2020)

---

> > ### Comment · Reviewer_vSCg · 2022-11-16
> > **Update on review**
> >
> > Thank you for the reply to my review, which includes various clarifications and a few changes in the revised version.
> >
> > The authors addressed all my minor concerns to my satisfaction. The remaining weakness of the paper, which was identified as such by all the reviewers, is the limitation to one specific instance of primal/dual algorithm. However, I agree with the authors that the included experiments are sufficient for demonstrating that the main idea is applicable and useful. The added discussion of potential candidates for future applications of the DAR scheme gives a nice picture of how future work can further investigate these ideas.
> > I appreciate the changes made to the manuscript, in addition, I would also recommend to clarify parts of the experimental setup in Section 4.2, as I do not seem to be the only reviewer to have struggled with understanding the details of the setup.
> >
> > Overall, after reading the other reviews and the author responses, I am still recommending acceptance of this work.

---

### Official Review · Reviewer_vEcN · 2022-10-25

**Confidence:** 3
**Correctness:** 4
**Technical Novelty And Significance:** 4
**Empirical Novelty And Significance:** 4
**Recommendation:** 8

**Clarity, Quality, Novelty And Reproducibility:**

This paper is clearly written except for one section: In Sec. 4.2, I didn't understand what Algorithm reconstruction does here. What is the expected output from the network and what is the training target and loss function of "learning steps of algorithm reconstruction"?
The idea of duality algorithmic reasoning is novel to my knowledge.
The paper presents sufficient details for reproducing results.

**Strength And Weaknesses:**

Strength
1 The idea of using duality for neural algorithmic reasoning is intuitive and interesting.
2 The implementation of DAR for max-flow and min-cut is technically sound.
3 Experiments on both synthetic data and real-world data are solid and demonstrate the advantage of neural algorithmic reasoning and dual algorithmic reasoning.

Weakness
The main concern is that this paper proposes only the idea of DAR and one demonstration of how to implement this idea for max-flow but lacks a general implementation method. It is unclear how difficult it is to set up a DAR model for other algorithms and tasks. Maybe proposing a general implementation is too difficult, more experiments on how to use duality for other algorithms are also welcome and helpful.

**Summary Of The Paper:**

This paper proposes DAR, an approach of Neural Algorithmic Reasoning following the FordFulkerson algorithm for calculating maximum flow. The key idea is to simultaneously learn the primal and the dual problem which is max-flow and min-cut in this case. This should produce a similar advantage to multi-task training. Experiments on synthetic graphs demonstrate that the proposed method performs better than learning to predict max-flow only. Experiments on real-world Brain Vessel Graphs demonstrate that DAR retains useful knowledge for extracting edge features that advance the performance of edge classification.

**Summary Of The Review:**

This is an interesting paper on neural algorithmic reasoning. The proposed approach is reasonable and achieves decent improvements compared with baselines. I lean toward accepting.

---

> ### Author Response · Authors · 2022-11-14
> **Reply to Reviewer vEcN**
>
> We thank the reviewer for their review and the kind words on our paper! We would like to address the reviewer concerns.
>
> First, we want to address the concerns (shared among all reviewers) for the lack of a general implementation. As the reviewer guesses, proposing a general implementation for any pair of primal/dual problems is indeed difficult. This is primarily a consequence of the fact that in neural algorithmic reasoning, in order to achieve OOD generalisation, there must provably be an alignment between the neural architecture and the dynamics of the target algorithm [1, 2]. This means that we might need to tweak the neural architecture for each different algorithm (if the dynamics are significantly different). In contrast, as long as the architecture is aligned, incorporating dual information should not be too hard. By our experiments, simply attaching a new _dual_ decoder works well.
>
> In fact, by looking at the experiment with the _pipeline_ architecture, we can see that it performs worse/on-par than the dual model. There, we engineered the architecture to try exploiting the dual information in a more complex way. In particular, we introduced a new processor whose sole goal was to learn min-cut, thus alleviating complexity on the other processors that could rely on the minimum cut being given. This did not result in improved performances, however. In light of these results, we believe that setting up a DAR model should be as simple as deploying an extra-decoder also for other pairs of primal/dual problems.
>
> As for _algorithm reconstruction_, we are happy to explain better in the following. Here, we have a neural reasoner $\phi$ that estimates flows in a capacitated graph. As per the encode-process-decode pipeline, such $\phi$ is defined as a composition of an encoder function $f$, a processor $p$, and a decoder function $g$, such that given some input graph $x$, the process can be described as $x \rightarrow_f z \rightarrow_p h \rightarrow_g y$. Notably, in this case $x$ contains the edge capacities $c$ among other inputs, i.e, $x = [..., c, …]$, and $f$ is learnt on this set of inputs (in other words, $f$ assumes the presence of $c$).
>
> The key point now is that we would like to re-use the knowledge in $\phi$ also on the new task of edge classification, which likely depends on the blood flow properties of the graph. However, as mentioned our encoding function $f$ assumes that the edge capacities are known but for the new BVG task we do not have access to the capacities $c$ as input features, i.e. let’s denote this as the input $x_{-c}$. What we do have access to instead are the three available BVG features: length $l$, distance $d$, and curvature $\rho$, i.e. $\tilde{x}  = [l, d, \rho, x_{-c}]$. Therefore, we’d like to learn how to estimate the flow on such inputs. In other words, we want to learn a map from this new input space $\tilde{x}$ to the output space of the Ford-Fulkerson algorithm.
>
> The only part of $\phi$ that really needs adapting to the new input space is the encoding function $f$, because $p$ and $g$ depend on latent representations instead. Thus, we replace $f$ with a new encoder $\tilde{f}$. Hence, what we call algorithm reconstruction is simply learning the new mapping $\tilde{x} \rightarrow_\tilde{f} z \rightarrow_p h \rightarrow_g y$ where $p$ and $g$ are kept frozen, and we only learn weights for $\tilde{f}$. In some sense we are learning to “reconstruct” the dynamics of the neural algorithm for the new inputs, i.e. algorithm reconstruction.
>
> We are open and happy to discuss further on this, should anything still be unclear. We hope this clarifies the related paragraph in the paper, should the reviewer have any suggestions whether to incorporate any of this in the paragraph, we will be happy to do as they suggest.
>
> [1] “What can neural networks reason about?” Keyulu Xu, Jingling Li, Mozhi Zhang, Simon S. Du, Ken-ichi Kawarabayashi, Stefanie Jegelka (ICLR 2020)
>
> [2] “How Neural Networks Extrapolate: From Feedforward to Graph Neural Networks”, Keyulu Xu, Mozhi Zhang, Jingling Li, Simon S. Du, Ken-ichi Kawarabayashi, Stefanie Jegelka (ICLR 2021)

---

### Author Response · Authors · 2022-11-14
**Summary of changes**

We thank the reviewer for their careful consideration and review of our manuscript. We are glad for the positive feedback and for the questions received.

We have introduced a number of changes that hopefully will address the reviewers’ questions and concerns. We report in the following a general summary of the changes and we will be providing more in-depth responses to each reviewer separately.

Summary of changes:

* In the conclusion, we included a short exposition of potential primal/dual candidates that might fit our dual algorithmic reasoning pipeline [Reviewer vEcN and vSCg].
* We updated Figure 1 including the missing residual connections [Reviewer vSCg].
* In the conclusion, we argue how max-flow/min-cut problems might be representative for a wide class of algorithms and problems for which strong duality holds (see also response to Reviewer 4SGE).
* We rephrased some claims in the conclusion about the generality of the empirical findings, making them more specific to max-flow/min-cut [Reviewer 4SGE].
* We added an additional experiment where we combined Node2Vec embeddings and dual embeddings (the best performing ones) for BVG data. This is intended to further verify that N2V and dual embeddings really carry two different types of information. We report these new results in Appendix C. We show that we can push even further the classification accuracy of classifiers using these combined pieces of information, suggesting that indeed dual embeddings encode a different type of information with respect to Node2Vec. Please check the response to Reviewer 4SGE.
* We added an extra section in the Appendix, namely Appendix D, where we report validation performance of all trained models, in order to better assess the OOD generalisation of table 1 in the main paper.

We are, of course, very happy to discuss further on any of the points raised during the remainder of the rebuttal period!

---

### Decision · Program_Chairs · 2023-01-20

**Decision:**

Accept: notable-top-25%

**Justification For Why Not Higher Score:**

It would be nice to see some rigorous theoretical analyses and experiments with more setups.

**Justification For Why Not Lower Score:**

The approach is novel and interesting.

**Metareview: Summary, Strengths And Weaknesses:**

This paper proposes a new approach to Neural Algorithmic Reasoning based on duality and presents experimental results to show its effectiveness. All reviewers found this idea novel. The AC agrees and recommends acceptance.

**Note From Pc:**

if the above contains the word "oral" or "spotlight" please see: "oral" presentation means -> notable-top-5% and "spotlight" means -> notable-top-25%. As stated in our emails, we are disassociating presentation type from AC recommendations